# G-SPARC: SPECTRAL ARCHITECTURES TACKLING THE COLD-START PROBLEM IN GRAPH LEARNING

## ABSTRACT

Graphs play a central role in modeling complex relationships across various domains. Most graph learning methods rely heavily on neighborhood information, raising the question of how to handle *cold-start nodes* — nodes with no known connections within the graph. These models often overlook the cold-start nodes, making them ineffective for real-world scenarios. To tackle this, we propose G-SPARC, a novel framework addressing cold-start nodes, that leverages generalizable spectral embedding. This framework enables extension to state-of-the-art methods making them suitable for practical applications. By utilizing a key idea of transitioning from graph representation to spectral representation, our approach is generalizable to cold-start nodes, capturing the global structure of the graph without relying on adjacency data. Experimental results demonstrate that our method outperforms existing models on cold-start nodes across various tasks like node classification, node clustering, and link prediction. G-SPARC provides a breakthrough built-in solution to the cold-start problem in graph learning. Our code will be publicly available upon acceptance.

## 1 INTRODUCTION

Graphs have advanced deep learning techniques across various domains, enabling tasks such as node classification, node clustering, and link prediction (Kipf & Welling, 2016a; Battaglia et al., 2018; LeCun et al., 1989; Yao et al., 2019; Chen et al., 2019). By incorporating both node features and structural relationships, graph-based models address complex relational patterns that traditional methods struggle to capture, positioning graph learning at the center of modern machine learning.

A major, yet often overlooked, challenge in graph learning is generalizing to nodes that emerge without initial connections. This scenario frequently occurs in real-world applications but is poorly addressed by existing methods. This issue, commonly referred to as the *cold-start* problem, is particularly crucial in dynamic environments where new nodes regularly appear without any links. For instance, on social media platforms, new users often join without any initial connections or followers. Despite having detailed profiles, these users do not have the connections that graph-based methods rely on to make predictions. As real-world graphs are not static; they are constantly evolving as new nodes may appear without connections, necessitating models that can adapt to these changes.

While state-of-the-art methods, such as message-passing (Gilmer et al., 2017; Wu et al., 2020), graph convolutional networks (Kipf & Welling, 2016b; Chiang et al., 2019), and graph transformers (Chen et al., 2022; Fu et al., 2024), excel on benchmark datasets, they fall short when applied to real-world scenarios that involve cold-start nodes. These models heavily rely on the neighborhood information, leaving a significant gap in their ability to make accurate predictions for cold-start nodes. This limitation represents a major obstacle to deploying graph-based solutions in practical settings.

To address the limitations of traditional graph learning methods, we leverage a fundamental concept in graph theory by transitioning from a graph representation defined by the adjacency matrix to its spectral representation captured through the eigenvectors of the Laplacian matrix. Spectral embedding represents the location of the node in the manifold coordinate system (Lafon et al., 2006; Belkin & Niyogi, 2003). We infer neighborhood from the spectral embedding for cold-start nodes, bypassing the need for explicit adjacency information.

Our approach uses a generalizable spectral embedding framework that provides spectral embedding for the cold-start nodes allowing isolated nodes to be seamlessly integrated into the graph structure. Our approach involves training a neural network to map node features to their corresponding spectral embeddings. During training, adjacency information is used to guide this mapping, but the model is generalizable to cold-start nodes during inference when no connections are available. This parametric mapping enables us to compute spectral embeddings that reflect the graph's underlying structure, even in the absence of adjacency data for the cold-start nodes.

In this paper, we propose a novel spectral-based framework specifically designed to handle cold-start node predictions across multiple downstream tasks. Rooted in spectral theory, our architectures leverage generalizable spectral embeddings to support a wide range of graph learning applications. We demonstrate the effectiveness of our framework across three key applications: node classification, node clustering, and link prediction. Our contributions address a critical gap in the current landscape of graph learning and offer a key extension for current methods to handle cold-start nodes making them suitable for real-world applications. Our framework is adaptable and can be seamlessly integrated into existing and future graph learning models to support cold-start nodes. Additionally, we show that our generalizable spectral embedding can be used for graph partitioning, enabling an effective mini-batching strategy that is well-suited for GCN methods.

## 2 RELATED WORK

In graph machine learning, most state-of-the-art algorithms assume the graph structure is fixed (e.g. Kipf & Welling (2016b); Veličković et al. (2017); Hamilton et al. (2017a); Xu et al. (2018); Gasteiger et al. (2018); Wu et al. (2019); Chiang et al. (2019); Chen et al. (2022); Thorpe et al. (2022); Mo et al. (2022); Liu et al. (2023)), thereby bypassing the cold-start problem. While these algorithms are effective on benchmark datasets, they are limited to real-world applications, where new cold-start nodes are common, making it crucial to develop models that can handle evolving graphs.

While the issue of cold-start nodes is often overlooked, few methods have been developed to address similar problems (Liu et al., 2020; 2021; Rong et al., 2019; Zhao et al., 2022; Hu et al., 2022), focusing on tail nodes - nodes with low-connectivity. A few methods specifically developed for handling cold-start scenarios include GraphSAGE (Hamilton et al., 2017a) and Cold-Brew (Zheng et al., 2021). Cold-Brew introduces a distillation technique where a trained "teacher" GCN model imparts knowledge to a "student" model, enabling the student to predict the low-dimensional embedding learned by the teacher. GraphSAGE learns a cluster-based representation of the graph by aggregating features from neighboring nodes to create an inductive representation. Unlike these methods that focus on representing the graph's structure, our approach learns a representation of the manifold from which the graph is drawn, utilizing spectral embeddings. Grounded in spectral graph theory, which is based on sound mathematical principles, these embeddings preserve both global and local structures, enabling the generation of meaningful representations for cold-start nodes while maintaining the overall topology. As a result, our method enhances the ability to capture the graph's structure, leading to improved performance and more accurate predictions in downstream tasks.

## 3 PRELIMINARIES

**Graphs.** An undirected graph $\mathcal{G} = (\mathcal{V}, \mathcal{E}, X)$ consists of a set of nodes $\mathcal{V}$, a set of edges $\mathcal{E}$, and $X$ the nodes features. The adjacency matrix $A$ of $\mathcal{G}$ has entries $A_{i,j} = 1$ if there is an edge between nodes $v_i$ and $v_j$, and 0 otherwise. The degree matrix $D$ is diagonal, with $D_{i,i} = \sum_{j=1}^{n} A_{i,j}$. An isolated node is a node with no known edges, i.e., $\forall v_j \in \mathcal{V}, (v_i, v_j) \notin \mathcal{E}$. The normalized graph Laplacian is defined as $L = I - D^{-1/2}AD^{-1/2}$.

**SpectralNet.** SpectralNet (Shaham et al., 2018) is a deep-learning approach designed for spectral clustering. It maps data points to the approximate $k$ eigenvectors of the Laplacian matrix, which is constructed from the similarities between the data points. By leveraging the convergence properties of the Laplacian operator applied in mini-batches (Belkin & Niyogi, 2001; 2003), SpectralNet enables efficient spectral clustering even for large-scale datasets by utilizing SGD to approximate the eigenvectors of the Laplacian. One key advantage of SpectralNet is that it learns a parametric mapping, approximating the eigenfunction of the Laplace-Beltrami operator. This allows a general-

ization to out-of-sample data (OOSE). This makes it possible to compute spectral embeddings for unseen data points. The learning process involves minimizing a Rayleigh-quotient loss function: trace $\left(Y^T L Y\right)$, s.t. $Y^T Y = I$, where $Y \in \mathbb{R}^{n \times k}$ represents the network outputs, and $L$ is the graph Laplacian. This method serves as the foundation for our generalizable spectral embedding, extending spectral techniques to settings of traditional graph-structured data.

**Spectral-GCNs.** Spectral Graph Convolutional Networks (GCNs) represent a significant advancement in graph-based machine learning by enabling convolution-like operations directly on graph-structured data. These networks adapt traditional convolutional operations from Euclidean domains to non-Euclidean spaces, such as graphs, making them highly effective for tasks involving relational data.

Spectral convolution (Kipf & Welling, 2016b) operates in the spectral domain of the graph, defining convolution as the multiplication of a signal $x \in \mathbb{R}^N$ by a filter $g_\theta = \text{diag}(\theta)$, parameterized by $\theta \in \mathbb{R}^N$, in the Fourier domain. Specifically, this operation can be expressed as applying the Fourier transform, followed by its inverse. In this context, $U$ is the matrix of eigenvectors of the graph Laplacian, which serves as the basis for the graph's Fourier transform. First, $x$ is transformed into the spectral domain using $U^T x$. Then, it is multiplied by $g_\theta$, a filter defined in the spectral domain. Finally, the inverse Fourier transform is applied by multiplying with $U$, transforming the filtered signal back into the original graph domain. The primary limitation of spectral GCNs is the need to compute the eigenvectors $U$ of the graph Laplacian. This requires the entire Laplacian matrix to be available, making the approach unsuitable for incorporating new nodes into the graph. Moreover, for cold-start nodes, recomputing the Laplacian eigenvectors is not only computationally expensive but also futile. Since the purpose of the Laplacian in GCNs is to enable convolution over connected nodes, nodes without any connections will not benefit from this operation. Making spectral-GCNs unsuitable for the generalization of cold-start nodes.

**Graph Transformers.** Transformers (Vaswani et al., 2017), originally developed for natural language processing, have been successfully adapted to handle graph data. A key development in this area is the Graph Transformer (Dwivedi & Bresson, 2020), which extends the transformer architecture to graph structures. This approach incorporates spectral embeddings for positional encoding, enabling the model to capture the relative positions of nodes within a graph.

However, standard transformer architectures face scalability issues when applied to large graphs due to the high computational complexity of the attention mechanism. This challenge has highlighted the need for more efficient methods that can handle large-scale graph data while maintaining the advantages of transformer-based models. State-of-the-art methods such as NAGphormer (Chen et al., 2022) and VCR-Graphormer (Fu et al., 2024) address this by creating token lists for each node, which aggregate key information from their neighborhoods. By focusing only on the most relevant data for each node, these approaches significantly reduce the computational load of the attention mechanism. While these methods excel in performance and scalability, they rely on adjacency information during inference, limiting their applicability in real-world environments where new nodes lack initial connections. Our work builds on these advancements by introducing a generalizable solution that supports cold-start nodes, enabling predictions without requiring adjacency information.

## 4 G-SPARC

In this section, we present the proposed G-SPARC framework in detail. To address the cold-start problem in graph learning, we begin by introducing our generalizable spectral embedding framework. Sections 4.2 and 4.3 then demonstrate how we adapt state-of-the-art architectures to handle cold-start nodes by integrating our generalizable spectral embeddings for the node classification task. Finally, we highlight additional applications of our approach for other graph-based downstream tasks.

### 4.1 SPARC - GENERALIZABLE EMBEDDING

It is reasonable to assume that graph data is drawn from an unknown manifold $\mathcal{M}$, with the graph $\mathcal{G}$ serving as a discrete approximation of this manifold. Spectral embeddings map the manifold $\mathcal{M}$ by computing the first $k$ eigenvectors of the graph's Laplacian matrix. These eigenvectors

$(u_1, u_2, \ldots, u_k)$, corresponding to the smallest eigenvalues, offer a low-dimensional representation of the graph. In this space, Euclidean distances approximate the diffusion distance of the underlying manifold $\mathcal{M}$ (Lafon et al., 2006).

Cold-start nodes—those without any existing connections in the graph—still lie on the manifold $\mathcal{M}$ despite their missing adjacency information. Let $v_{\text{cold}}$ represent a cold-start node. While identifying neighbors based solely on the graph structure is impossible for such nodes, one straightforward approach is to estimate their neighborhood purely by feature similarity. However, this naive method fails to capture the intrinsic geometry of the graph. As shown in Section 5.1, the feature space does not naturally align with the graph's manifold, resulting in suboptimal performance.

We address the cold-start challenge by introducing a parametric mapping, $\mathcal{F}_\theta : \mathbb{R}^d \to \mathbb{R}^k$, which approximates the first $k$ eigenfunctions of the Laplace-Beltrami operator, thereby forming the spectral embeddings of the nodes. This mapping takes node features as input and projects cold-start nodes onto the graph's manifold, enabling the identification of relevant neighbors through Euclidean distance in the spectral embedding space.

Crucially, the graph structure is only used during training. This enables the learned spectral embedding to generalize to cold-start nodes. During inference, for cold-start nodes with only node features, we compute approximate spectral embeddings. This allows us to identify their nearest neighbors in the graph spectral space (see Figure 1). This generalizable spectral embedding framework enables cold-start nodes to be integrated into various downstream tasks, such as node classification and clustering, even in the absence of adjacency information. In this paper, we incorporate our generalizable spectral embedding into two state-of-the-art architectures for node classification and demonstrate its straightforward application to link prediction and node clustering. This approach enables accurate predictions for cold-start nodes across these tasks, even in the absence of adjacency information.

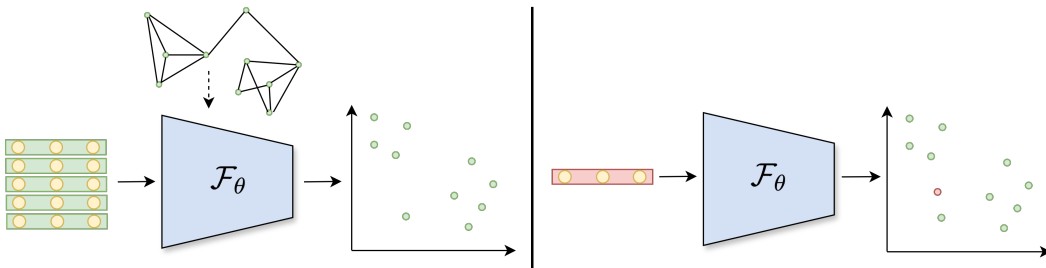

Figure 1: The pipeline of our SPARC embedding model for learning spectral embeddings in both training and inference phases. *Left:* During training, the input consists of node features, and the model outputs the spectral embedding of these nodes. The model uses the graph structure through the Laplacian matrix, which is incorporated into the loss function during the training (see loss function of Section 3). The model maps nodes close in the graph to also be close in the embedding, effectively transitioning the graph from a discrete to a continuous space. *Right:* During inference, for cold-start nodes, the input is only the node's features. The trained model generates the approximate spectral embedding of the cold-start node, allowing us to infer its neighborhood and relationships based solely on node features.

## 4.2 SPARC-GCN

Spectral-GCN (Kipf & Welling, 2016b) applies graph convolutional networks in the spectral domain, utilizing spectral embeddings to perform convolution-like operations on graph-structured data. The convolution is formally defined as:

$$X_{l+1} = U g_\phi U^T X_l$$

Where $U$ denotes the matrix of eigenvectors of the graph Laplacian, $g_\phi$ is the spectral filter, and $X$ represents the nodes' features. This method enables effective feature aggregation across the graph.

A significant limitation arises for cold-start nodes. For a cold-start node $v$ with features $x$, traditional convolution becomes impossible. Furthermore, the computing of $U$ with cold start nodes is futile as the convolution applies only to connected nodes.

To overcome these limitations, we propose a modification to the Spectral-GCN architecture. We replace the spectral embedding $U$ with the output of a parametric map $\mathcal{F}_\theta$ denoted as $U_\theta$, a function on the feature space. Our modified convolution is defined as:

$$X_{l+1} = U_\theta(X)g_\phi U_\theta(X)^T X_l$$

Where $U_\theta(X)$ is the approximated spectral embedding using features $X$, and $g_\phi$ is a learnable diagonal matrix. This modification preserves the advantages of spectral-GCN while enhancing practicality for deployment during inference.

For a cold-start node $v$ with features $x$ but no connections, we: (1) Find the spectral embedding of the cold-start node, denoted as $U_\theta(x)$. (2) Perform convolution over the new set $\hat{X} = [X; x]$ that includes the new node. The node is processed using both its features and the spectral embedding: $x_{l+1} = U_\theta(x)g_\phi U_\theta(\hat{X})^T \hat{X}_l$. This approach effectively eliminates the need for adjacencies from cold-start nodes, enabling predictions through convolution on such nodes.

To handle larger graphs where training in a single batch is infeasible, we developed a mini-batching strategy for spectral GCNs. This approach, detailed in Section 4.4, enables scalable training by partitioning the graph into manageable sub-graphs.

### 4.3 SPARCPHORMER

NAGphormer (Chen et al., 2022), a scalable graph transformer architecture, introduces a novel approach in which each node is assigned a token list, constructed by aggregating features from its neighbors. Self-attention is then applied exclusively within this token list, rather than across all node pairs, capturing dependencies and relationships between the nearest neighbors. This approach significantly improves scalability and enables efficient training of graph transformers.

$$\mathcal{T}(v_i) = [h_0(v_i), h_1(v_i), \ldots, h_k(v_i)]$$

$$h_j(v_i) = \sum_{v \in \mathcal{V}} X[v] \cdot \mathbf{1}\left(\text{dist}(v, v_i) <= j\right)$$

The token list $\mathcal{T}(v_i)$ for node $v_i$ consists of aggregated feature representations $h_j(v_i)$ for each node $v$ that is exactly $j$ hops away from $v_i$. Here, $X[v]$ represents the feature vector of node $v$, and $\mathbf{1}\left(\text{dist}(v, v_i) = j\right)$ is an indicator function that returns 1 if node $v$ is within $j$ hops away from $v_i$, and 0 otherwise. The sum is computed over all nodes $v$ in the graph. This construction enables neighborhood aggregation over multiple hops in the graph.

However, NAGphormer is not suitable for cold-start nodes, as it relies on adjacency information, which is unavailable for nodes without initial connections.

Our model extends the NAGphormer architecture by incorporating generalizable spectral embeddings, allowing us to address the cold-start problem. We modify the token list creation process by using the node features of the nearest neighbors in the spectral embedding space (measured by Euclidean distance), instead of relying on adjacency information.

$$\mathcal{T}(v_i) = [g_0(v_i), g_1(v_i), \ldots, g_k(v_i)]$$

$$g_j(v_i) = \sum_{v \in \mathcal{V}} X[v] \cdot \mathbf{1}\left(\|\mathcal{F}_\theta(v_i) - \mathcal{F}_\theta(v)\|_2 \text{ is within the nearest } 2^j \text{ neighbors in } \mathcal{F}_\theta(X)\right)$$

The token list $\mathcal{T}(v_i)$ for node $v_i$ is constructed from aggregated feature vectors $g_j(v_i)$ based on the nearest neighbors in the spectral embedding space. Here, $g_j(v_i)$ aggregates the features $X[v]$ of nodes $v$ that are among the $2^j$ closest nodes to $v_i$ in the spectral embedding space, where the distance is measured by Euclidean distance $\|\mathcal{F}_\theta(v_i) - \mathcal{F}_\theta(v)\|_2$. The mapping $\mathcal{F}_\theta$ is parameterized by a neural network, and $\mathbf{1}(\cdot)$ is an indicator function.

By eliminating the dependency on adjacency information, we enable predictions for cold-start nodes, effectively overcoming the limitations of previous graph-based transformers. Utilizing node features

in the spectral embedding space allows the model to focus on the most relevant neighbors while maintaining low computational complexity. Applying self-attention in this way ensures accurate predictions for new nodes and enhances scalability and robustness, extending the applicability of graph transformers to cold-start scenarios.

## 4.4 ADDITIONAL APPLICATIONS

**Clustering and Mini-Batching.** When applying Euclidean methods like k-means on spectral embeddings to partition the graph $\mathcal{C} = k\text{-means}(\mathcal{F}_\theta(X), k)$ —a process known as spectral clustering—nodes are partitioned into $k$ clusters with small intra-cluster diffusion distances and large inter-cluster distances. This technique effectively partitions the graph into sub-graphs, which offer valuable insights into the graph's structure. The partitioning retains essential, close connections between nodes, while minimizing the loss of less significant, distant connections, making it ideal for tasks that require dividing the graph into meaningful substructures while preserving as much information as possible.

We found that spectral partitioning is particularly useful for mini-batching in GCNs, where maintaining local neighborhoods and graph structure during training is essential. GCNs rely on aggregating information from neighboring nodes within each batch, and splitting important connections across different mini-batches can hinder the network's ability to learn meaningful node representations. Spectral partitioning overcomes this issue by clustering highly connected nodes into the same mini-batch, preserving local structural information and improving the efficiency of the learning process.

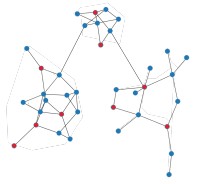 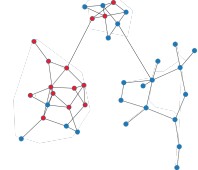 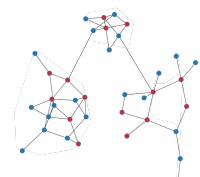 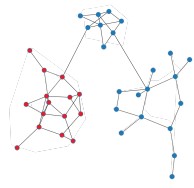

(a) Random Selection    (b) 2-Hop Neighbor Selection    (c) ClusterGCN Selection    (d) SPARC Clustering Selection

Figure 2: Visualization of node sampling using common selection methods and our proposed spectral clustering method. Red indicates nodes selected for the batch. (a) shows random node selection, where meaningful convolution is unlikely due to the disconnection between nodes.(b) select random node along with its hop-1 and hop-2 neighbors, ensuring connectivity between nodes in the batch, but potentially from an irrelevant cluster. (c) depicts the node selection using the ClusterGCN partitioning, where each batch includes small, connected node groups. (d) demonstrates our spectral partition method, optimized for convolution, ensuring that each node interacts with other nodes in the same cluster, enhancing the effectiveness of convolution.

This approach also enhances computational efficiency by reducing the need for information exchange across batches. As a result, spectral partition in mini-batching allows GCNs to scale more effectively to larger graphs while maintaining strong predictive performance by preserving essential connectivity within each batch. In Figure 2, we illustrate the challenges of node sampling methods that are not based on diffusion distance, where nodes can extend beyond their cluster boundaries in the graph.

**Link Prediction.** Using the parametric map $\mathcal{F}_\theta$ has the capability to reconstruct the graph connections for cold-start nodes effortlessly. By simply ranking the Euclidean distance from the embedding of cold-start node $u_\theta = \mathcal{F}_\theta(x)$ to the top $r$ nodes in rank $R$, where $d(u_\theta, u_{R_1}) \leq d(u_\theta, u_{R_2}) \leq \ldots \leq d(u_\theta, u_{R_n})$ in the spectral embedding. Notably, the embedding is not trained to the specific task of link prediction but our single embeddings apply for various tasks.

## 5 EXPERIMENTS

**Cold-Start Nodes.** A subset of nodes from the graph is isolated by masking their adjacency connections. Test nodes are defined as nodes that were not included in the training phase but still retain full adjacency information.

**Datasets.** The statistical properties of all datasets are summarized in Table 1. The Cora (Baum et al., 1972), Citeseer (Giles et al., 1998), and Pubmed (Sen et al., 2008) datasets are citation network datasets where nodes represent documents and edges represent citation links. For the Cora and Citeseer datasets, node features are represented using a bag-of-words

Table 1: Data statistics.

| Dataset | Nodes | Edges | Classes | Features |
|---------|-------|-------|---------|----------|
| Cora | 2,708 | 5,429 | 7 | 1,433 |
| Citeseer | 3,312 | 4,732 | 6 | 3,703 |
| Pubmed | 19,717 | 44,338 | 3 | 500 |
| Reddit | 232,965 | 11,606,919 | 41 | 602 |

approach. In contrast, the Pubmed dataset uses Term Frequency-Inverse Document Frequency (TF-IDF) values to represent node features. The Reddit dataset (Hamilton et al., 2017b) consists of posts from September 2014, with nodes representing individual posts and edges indicating interactions between posts by the same user.

### 5.1 RESULTS

**Cold-Start Classification.** The results in Table 2 summarize the node classification performance over the cold-start set, which consists of nodes without connections.

While the overall accuracy on test node remains comparable to traditional methods like Spectral-GCN and NAGphormer (see Appendix D), our approach uniquely enables accurate classification of cold-start nodes, effectively addressing the isolated node problem. The results for these architectures are not included, as they are not designed to handle cold-start nodes. Moreover, although methods like Cold-BREW [1] and GraphSAGE can handle isolated nodes, they exhibit a significant drop in accuracy compared to our approach. This underscores the strength of our method, which adapts state-of-the-art architectures to cold-start nodes by leveraging generalizable spectral embeddings and capturing the graph's global structural information, resulting in more effective classification of cold-start nodes.

Our proposed methods, SPARC-GCN and SPARCphormer, demonstrate competitive results. The variation in performance across datasets can be attributed to the inherent strengths of each method, as neither Spectral-GCN nor NAGphormer consistently outperforms the other in all scenarios, highlighting the need for adaptable approaches depending on the dataset characteristics. We evaluate classification accuracy across different datasets, with the accuracy on isolated nodes serving as a key indicator of our framework's effectiveness in addressing the cold-start problem.

Table 2: Classification accuracy of cold-start nodes

| METHOD | Cora | Citeseer | Pubmed | Reddit |
|--------|------|----------|--------|--------|
| G-SAGE | 66.02 ± 1.18 | 51.46 ± 1.30 | 69.87 ± 1.10 | 85.63 ± 0.66 |
| C-BREW | 68.92 ± 1.13 | 53.13 ± 0.24 | 72.32 ± 0.87 | OOM |
| SPARC-GCN | **73.88 ± 6.27** | 64.90 ± 3.19 | 82.78 ± 2.05 | **91.46 ± 0.92** |
| SPARCphormer | 68.49 ± 0.89 | **66.35 ± 0.92** | **84.66 ± 0.25** | 74.92 ± 0.23 |

**Cold-Start Clustering.** The results of node clustering are presented in Figure 3 for both connected nodes and cold-start nodes. Our proposed method, SPARC-Clustering, matches the performance of state-of-the-art techniques on the connected set while demonstrating superior effectiveness for cold-start nodes. Other methods considered include k-means, which disregards the graph structure; spectral clustering; and SSGC (Zhu & Koniusz, 2021), which utilize spectral methods but are unsuitable

---

[1]Code lacks support for large datasets, causing memory issues.

for cold-start scenarios. Additionally, R-GAE (Mrabah et al., 2022) employs an auto-encoder that is more appropriate for cold-start nodes; however, its loss calculation relies on Euclidean clustering methods that may be ill-fitting when graph features are non-convex. Our method is both generalizable for cold-start nodes and capable of clustering even when the features are non-convex. We measured the clustering accuracy for both connected and cold-start nodes.

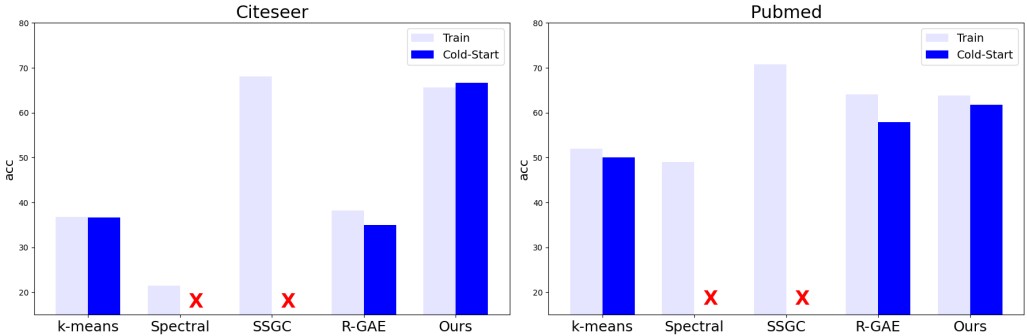

Figure 3: Clustering accuracy for train-connected nodes (light blue) and new cold-start nodes (blue) is evaluated. While spectral models yield adequate performance on the graph, they are unsuitable for cold-start scenarios. In contrast, Euclidean methods generalize better but perform well only when the graph features show convexity.

**Mini-Batching.** We experimented with training a simple GCN ($Y = \sigma(AXW)$) in two mini-batches, in the first experiment having a random partition over the dataset. In the second experiment training the same model with our mini-batching method. Random partition

Table 3: GCN classification accuracy

| METHOD | Cora | Citeseer | Pubmed |
|---|---|---|---|
| Random-Batches | 73.45 ± 5.44 | 61.33 ± 3.55 | 75.90 ± 3.93 |
| SPARC-Batches | 84.31 ± 1.24 | 70.57 ± 1.39 | 81.64 ± 1.56 |
| % Gain | 14.77% | 15.08% | 7.56% |

versus SPARC partition of the graph (trained on mini-batch SGD) results are shown in Table 3. Our spectral clustering partition leads to better performance since it removes fewer between-partition links. Measuring node classification accuracy in both settings.

In Figure 4 we show an experiment training Cluster-GCN (Chiang et al., 2019) with three mini-batching partitions for node classifaction. The mini-batching is by assigning each node a cluster out of 1500 and training 20 clusters in each batch. First having a random assignment. Second, using the ClusterGCN proposed method with METIS (Karypis & Kumar, 1997) partition, an algorithm that creates a tree out of the graph and parting the graph to sub-graphs of the immediate neighbors. Third, spectral clustering using SPARC over the Reddit graph dataset and using these clusters as the partition. We can see faster convergence using our method of training the same model.

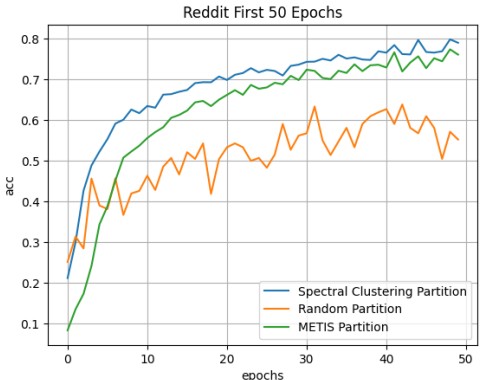

Figure 4: Our spectral partition versus METIS and random mini-batching partitions. The spectral clustered mini-batches convergence is faster at the beginning of the training process

**Cold-Start Link Prediction.** The results of link prediction are presented in Table 4. Link prediction for cold-start nodes is not well-researched; therefore, we compare our method to two approaches outlined in (Guo et al., 2023). LLP utilizes relational knowledge distillation and cross-modeling of two networks: MLP and GNN.

Table 4: Link Prediction

| METHOD | Cora | Citeseer | Pubmed |
|--------|------|----------|--------|
| LLP-MLP | 22.90 ± 2.22 | 28.21 ± 3.75 | 38.01 ± 1.67 |
| LLP | 27.87 ± 1.24 | 34.05 ± 2.45 | 50.48 ± 1.52 |
| Ours | 31.25 ± 2.63 | 34.25 ± 2.78 | 48.07 ± 1.50 |

While our approach achieves comparable results to state-of-the-art methods in the link prediction task, it is important to highlight that this performance is a natural extension of our generalizable spectral embedding method. Unlike more specialized models designed specifically for link prediction, our method does not require any additional architectural modifications or complex adjustments. The ability to perform link prediction comes inherently from the core spectral embedding framework, making it a byproduct of our main approach. In this sense, the comparable results we obtain are achieved with a simpler and more generalizable architecture, demonstrating the versatility and robustness of our method without requiring further optimizations specifically for link prediction.

We measure cold-start link prediction by computing the mean reciprocal rank (MRR). We rank the closest nodes in our embedding to the cold-start node and calculate the intersections in the top 20 nodes.

### LIMITATIONS OF FEATURE-BASED SIMILARITIES

Most graph-based methods utilize both node features and graph structure to make node-level predictions. The SPARC embedding, developed to address cold-start nodes, approximates the eigenvectors of the graph Laplacian captures only the graph structure. However, incorporating node affinities, denoted as $W$, introduces a distinct form of a Laplacian that for some tasks may be beneficial.

Similarity solely on the feature space in graphs often fails to capture the true underlying structure, leading to misleading results when used for graph analysis tasks. While node features can reflect shared characteristics or common properties, they do not necessarily correspond to the actual connectivity patterns within the graph. This arises because features may be unrelated to the graph topology, resulting in nodes with high feature similarity that are not directly linked or nodes that are closely connected but have divergent features. Consequently, relying on feature similarities alone can obscure critical structural information, undermining the effectiveness of algorithms that depend on understanding the graph's inherent organization. Therefore, integrating structural information is essential to accurately capture the relationships and dependencies in graph-based models.

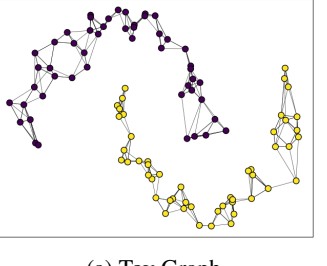
(a) Toy Graph

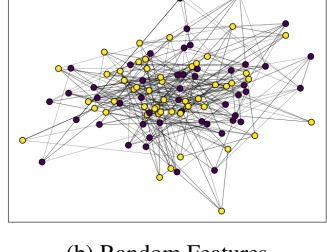
(b) Random Features

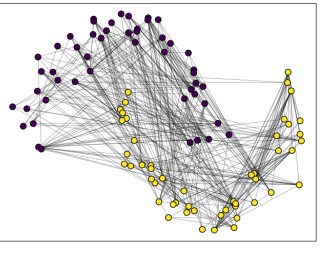
(c) Random Edges

Figure 5: (a) Toy non-convex dataset with two distinct clusters, converted into a graph by connecting each node to its $4$ nearest neighbors. (b) The same graph with randomized features—models using the adjacency matrix $A$ will yield identical predictions as in (a). (c) Randomized edges with original features—models using an affinity matrix on the features $W$ will still produce the same predictions as (a). Suggesting that real-world graphs are likely within this spectrum.

Based on the visualizations shown in figure 5, it is evident that different Laplacian matrices can offer advantages depending on the graph structure and the relationship between node features and connectivity. For example, as depicted in Figure 5c, when node features significantly influence

the graph structure, utilizing a Features-Edges View Laplacian matrix (see Appendix C), which integrates both feature similarity and adjacency information, can be advantageous.

We quantify this relationship through what we call the "Feature Weighting Factor", which assesses the extent to which node features correspond to the structural connectivity. This ratio guides the selection of the Laplacian type—balancing between feature-driven and adjacency-driven representation. It is crucial to note, however, that feature similarity alone does not guarantee an accurate reflection of the underlying graph structure, which underscores the importance of choosing the right Laplacian framework based on the specific characteristics of the graph and its features.

**Results.** The plot in Figure 6 showcases how the accuracy changes with different values of $\alpha$ feature weight, where $\alpha$ balances the contribution of features and edges in the Laplacian matrix. Notably, using only features ($\alpha = 1$) results in a decrease in performance, indicating that while features provide valuable information, they are insufficient on their own for optimal graph structure representation. However, a combination of features and edges ($\alpha$ values between 0 and 1) generally leads to improved performance, with an optimal $\alpha$ varying by dataset.

These observations underscore the delicate balance required in configuring the Laplacian matrix to harness both the graph structure and graph features effectively. While node features alone do not capture the complete picture, their integration with structural data at an optimal level significantly enhances performance.

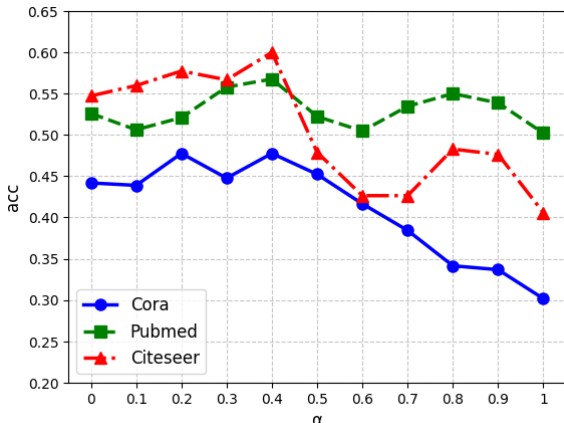

Figure 6: Clustering accuracy, $\alpha$ controls the ratio between adjacency and affinity matrix. Feature affinities enhance the embedding's performance in the clustering task; however, using features alone is insufficient.

## 6 CONCLUSION

In this work, we introduced G-SPARC, a novel spectral-based method designed to address the cold-start problem in graph learning. Our approach effectively integrates cold-start nodes into state-of-the-art graph learning methods, enabling accurate predictions for cold-start nodes. Experimental results demonstrate that G-SPARC outperforms existing models in handling cold-start nodes, providing a solution for real-world applications where new nodes frequently appear. The adaptability of our method allows it to be seamlessly integrated into existing and future graph learning frameworks, enhancing their capability to manage evolving graphs. In this paper, we introduced two novel adaptations, SPARC-GCN and SPARCphormer, to existing state-of-the-art methods, enabling them to effectively manage cold-start nodes and making them suitable for real-world applications.

A limitation of our method is its dependency on the meaningful node features. G-SPARC may not perform optimally if the node features are random or lack any relationship with the graph's manifold. However, it is generally safe to assume that in practical scenarios, the features will have some correlation with the graph structure, which supports the applicability of our method.

Our research has primarily focused on the common real-world scenario of homophilous graphs, where the graph structure plays a crucial role in node-level predictions. We also explored the feature weighting factor, which becomes particularly significant in heterophilous graphs where node features drive the predictions. Moving forward, we plan to continue our research to include heterophilous graphs, aiming to adapt and refine our approaches.

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

## A  DATASETS

Table 5: Characteristics of the datasets in our experiments

| Dataset | Nodes | Edges | Classes | Feats | Avg Deg | Cold-Start |
|---------|-------|-------|---------|-------|---------|------------|
| Cora | 2,708 | 5,429 | 7 | 1,433 | 4.89 | 3% |
| Citeseer | 3,312 | 4,732 | 6 | 3,703 | 3.77 | 3% |
| Pubmed | 19,717 | 44,338 | 3 | 500 | 5.49 | 3% |
| Reddit | 232,965 | 11,606,919 | 41 | 602 | 492 | 3% |

## B  SPARC PRAMETRIC MAP

The generalization process borrows key ideas from SpectralNet (Shaham et al., 2018) and spectral clustering to achieve a scalable and generalizable method for the first $k$ eigenvectors of the graph Laplacian. A key idea in spectral clustering is that embedding of the first $k$ eigenvectors (where $k << n$) captures the most significant variations in the graph structure.

We computes the Laplacian matrix for a mini-batch using the graph adjacencies to find the parametric map using a neural network with orthogonal enforcement in the last layer. The training process is in a coordinated descent fashion, where we alternate between orthogonalization and gradient steps. Each of these steps uses a different mini-batch (possibly of different sizes), sampled from the training set $X$.

To learn the spectral embeddings of the graph, we use the following Rayleigh-quotient loss:

$$\mathcal{L}_{RQ} = \text{trace}\left(Y^T L Y\right) \quad \text{s.t.} \quad Y^T Y = I \tag{1}$$

Where $Y$ is the network output and $L$ is the sub-Laplacian. A rotation and reflection ambiguity of the loss refrains a straight transformation to create a spectral embedding. Such rotation can be eliminated by computing the eigendecomposition of the estimated eigenvalues on $k \times k$ matrix (Anonymous, 2024)).

The map learning process is detailed in Algorithm 1 and visually represented in Figure 7. The resulting embeddings correspond to the top $k$ eigenvectors of the Laplacian matrix. Furthermore, the model is optimized to handle future nodes without incorporating any edge information during training.

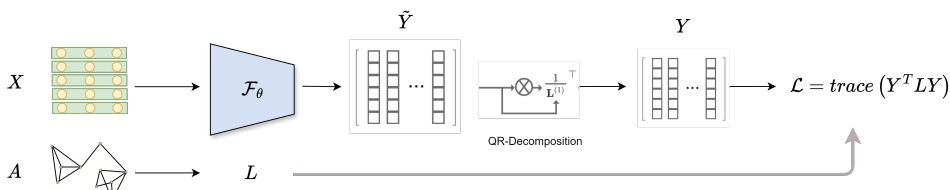

Figure 7: Learning a parametric map from the feature space $X$ to the first $k$ vectors of the Laplacian and undergoes QR decomposition to ensure orthogonality

---

**Algorithm 1** $\mathcal{F}_\theta$ Training

---

**Require:** $X \in \mathbb{R}^d$, number of vectors $k$, batch size $m$
**Ensure:** Embedding $y_1, \ldots, y_n \in \mathbb{R}^k$
 1: Part $X$ to mini-batches of size $m$ with neighbors
 2: Randomly initialize the network weights $\theta$
 3: **while** $\mathcal{L}_{RQ}$ not converged **do**
 4:     Orthogonalization step:
 5:         Sample a mini-batch $X$ of size $m$ and the corresponding $A$ size $m \times m$;
 6:         Forward propagate $X$ and compute inputs to orthogonalization layer $\hat{Y}$
 7:         Compute the QR factorization $LL^T = \hat{Y}^T \hat{Y}$
 8:         Set the weights of the orthogonalization layer to be $\sqrt{m}(L^{-1})^T$
 9:     Gradient step:
10:         Sample a mini-batch $X$ of size $m$ and the corresponding $A$;
11:         Forward propagate $x_1, \ldots, x_m$ to get $y_1, \ldots, y_m$
12:         Compute the loss $\mathcal{L}_{RQ}$
13:         Use the gradient of $\mathcal{L}_{RQ}$ to tune all $\mathcal{F}$ weights, except those of the output layer
14: **end while**
15: Forward propagate $x_1, \ldots, x_n$ and obtain $\mathcal{F}_\theta$ outputs $y_1, \ldots, y_n$

---

## C   EXPLORING DIFFERENT LAPLACIANS

Spectral eigendecompositions are typically performed solely on the graph structure, oblivious to any additional information associated with the graph. However, in most deep-learning models, different Laplacian matrices can yield better results for specific tasks to enhance unsupervised spectral embeddings. In our study, we explored two key approaches:

**K-Power Random Walk.**   The k-power random walk method captures context from the k-hop neighbors within a graph by leveraging the k-power of the normalized adjacency matrix. The key idea lies in summing up these normalized matrices for each step, resulting in a new matrix denoted as $A_{\text{k-power}} = \sum_i^k \text{normalized}(A^i)$, where $A$ is the graph adjacency matrix and normalization defined as $D^{-1/2}AD^{-1/2}$. This operation effectively simulates multiple convolutions, leading to smoother graphs with tighter clusters. In essence, the k-power random walk bridges local and global information, enhancing the expressive power of the graph.

Figure 8b evaluates the effect of increasing the power $k$ of the adjacency matrix, which integrates increasingly distant neighborhood information into the graph representation. Initially, as $k$ increases, performance improves, reflecting the benefits of incorporating broader contextual information. However, beyond a certain point, further increases in $k$ lead to performance degradation, evident from the sharp declines for all datasets at higher $k$ values. This phenomenon, known as oversmoothing, occurs because the node representations begin to lose their distinctive characteristics, converging to a similar state that dilutes the useful signals for the learning tasks.

**Features-Edges View.**   In graph-based data, we construct simulated graphs with both node features and connections. In the context of node clustering, nodes may have defining features that indicate

labels, neighborhood-defining labels, or a combination of both. Adapting ideas from traditional spectral clustering methods of constructing an affinity matrix based on the nodes' features. Given $n$ data points, an affinity matrix $W$ is an $n \times n$ matrix whose $W_{i,j}$ entry represents the similarity between $x_i$ and $x_j$. A popular choice for $W$ is the Gaussian kernel: $W_{i,j} = \exp\left(-\frac{\|x_i - x_j\|^2}{2\sigma^2}\right)$ where $\sigma$ is a defined bandwidth. We construct the Laplacian matrix for a linear combination of the original adjacency matrix $A$ and the affinity matrix $W$: $A_{\text{feat-edge}} = \alpha * W + (1 - \alpha) * A$, where $\alpha$ depends on the characteristic of the nodes' features.

The results are presented in Section 5.1.

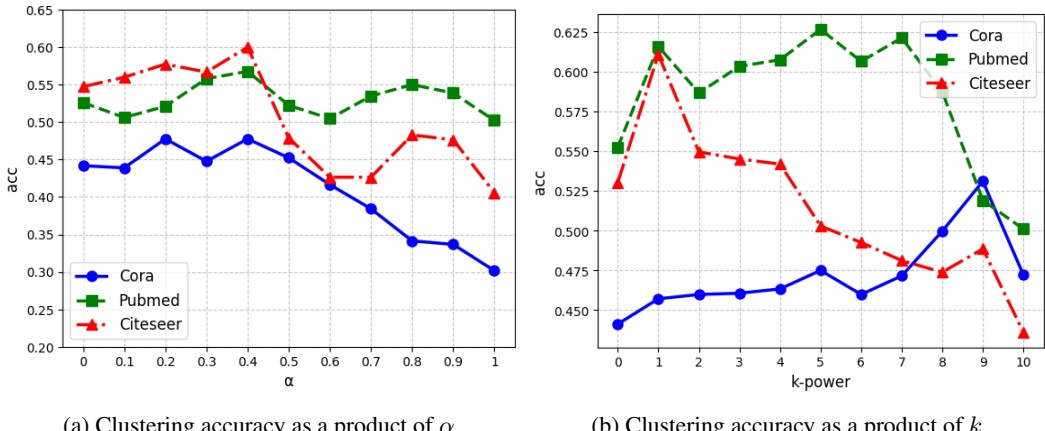

(a) Clustering accuracy as a product of $\alpha$    (b) Clustering accuracy as a product of $k$

Figure 8: Visualization of node clustering demonstrating the effects of over-smoothing and the effect of using solely feature affinities or adjacencies.

## D  ADDITIONAL RESULTS

We present classification accuracy for two scenarios: 'Test', where nodes have full adjacency information, and 'Cold-Start', where connectivity data is missing. Our results show competitive performance with state-of-the-art models in the 'Test' scenario, while significantly outperforming existing methods in handling cold-start nodes. This highlights our method's unique capability to extend graph learning applications to effectively manage isolated nodes.

Table 6: Classification

| METHOD | Cora | | Citeseer | | Pubmed | | Reddit | |
|---|---|---|---|---|---|---|---|---|
| | Test | Cold Start | Test | Cold Start | Test | Cold Start | Test | Cold Start |
| Spectral-GCN | 87.94 ± 0.85 | NS | 77.92 ± 0.61 | NS | 86.20 ± 0.41 | NS | OOM | NS |
| NAGphoremer | 89.55 ± 0.48 | NS | 76.32 ± 0.52 | NS | 88.30 ± 0.29 | NS | 93.75 ± 0.03 | NS |
| G-SAGE | 83.92 ± 1.25 | 66.02 ± 1.18 | 71.78 ± 2.67 | 51.46 ± 1.30 | 82.16 ± 1.92 | 69.87 ± 1.10 | 94.32 ± 0.00 | 85.63 ± 0.66 |
| C-BREW | 84.66 ± 0.00 | 69.62 ± 0.00 | 71.18 ± 0.00 | 53.17 ± 0.00 | 86.81 ± 0.00 | 72.33 ± 0.00 | OOM | OOM |
| SPARC-GCN | 84.46 ± 1.51 | **73.88 ± 6.27** | 65.44 ± 4.23 | 63.66 ± 0.00 | 86.46 ± 3.77 | 82.78 ± 2.05 | 93.04 ± 0.87 | **91.46 ± 0.92** |
| SPARCphormer | 69.41 ± 1.65 | 68.49 ± 0.89 | 70.87 ± 0.43 | **66.35 ± 0.92** | 85.12 ± 0.46 | **84.66 ± 0.25** | 78.95 ± 0.76 | 74.92 ± 0.23 |

# E    ALGORITHMS

---

**Algorithm 2** SPARC-GCN Training

---

**Require:** Node features $X \in \mathbb{R}^{n \times d}$, batch size $m$, labels $\mathbf{y}$ for training nodes
**Ensure:** Predicted labels $\hat{\mathbf{y}}_1, \ldots, \hat{\mathbf{y}}_n$
 1: Compute spectral embeddings $U$ for all nodes in $X$ (generalizable to cold-start nodes)
 2: Randomly initialize model parameters $\theta$
 3: **while** $L_{\text{Spectral-GCN}}(\theta)$ not converged **do**
 4:      Sample a mini-batch of near-neighbors in the spectral space $U$
 5:      Forward propagate Specral-GCN-Layers and Linear Layers get final embeddings $z_1, \ldots, z_m$

 6:      Apply a linear layer to predict labels $\hat{\mathbf{y}}_1, \ldots, \hat{\mathbf{y}}_m$
 7:      Compute the loss $L_{\text{Spectral-GCN}}(\theta)$ using the ground-truth labels $\mathbf{y}$
 8:      Update the model parameters $\theta$ using gradient descent
 9: **end while**
10: **Cold Start Inference:** $\hat{x}$
11: Predict $\hat{U}$ spectral embedding of the cold start node
12: Identify the $k$ nearest neighbors of $\hat{x}$ in the spectral embedding space $U$
13: Forward propagating the new inference batch

---

**Algorithm 3** SPARCphormer Training

---

**Require:** Node features $X \in \mathbb{R}^{n \times d}$, number of neighbors $k$, labels $\mathbf{y}$ for training nodes
**Ensure:** Predicted labels $\hat{\mathbf{y}}_1, \ldots, \hat{\mathbf{y}}_n$
 1: Compute our SPARC embeddings for all nodes in $X$
 2: **for** each node $v \in \mathcal{V}$ **do**
 3:      Identify the $2^k$ nearest neighbors of $v$ in the spectral embedding space
 4:      Construct token list for $v$ using features of the $2^k$ nearest neighbors as described in 4.3
 5: **end for**
 6: Randomly initialize model parameters $\theta$
 7: **while** $\mathcal{L}_{\text{nll}}(\theta)$ not converged **do**
 8:      Sample a mini-batch of training nodes and their token lists
 9:      Forward propagate token lists through the self-attention mechanism to get final embeddings
         $z_1, \ldots, z_m$
10:      Apply a linear layer to predict labels $\hat{\mathbf{y}}_1, \ldots, \hat{\mathbf{y}}_m$
11:      Compute the loss $\mathcal{L}_{\text{nll}}(\theta)$ using the ground-truth labels $\mathbf{y}$
12:      Update the model parameters $\theta$ using gradient descent
13: **end while**

---

# F    TECHNICAL DETAIL AND HYPER-PARAMETERS.

For fairness, we run each of the compared algorithms ten times on the above datasets, recording both the mean and standard deviation of their performance. The same backbones are employed across all methods and datasets. All external algorithms provided hyper-parameters and so each run consists of the reported parameters.

Table 7: Hyperparameters for SPARC-Embeddings

| Parameter | Cora | Citeseer | Pubmed | Reddit |
|---|---|---|---|---|
| Hidden Dimension | 512, 256, 32 | 512, 256, 32 | 512, 256, 32 | 512, 256, 64 |
| K eigenvectors | 32 | 32 | 32 | 64 |
| Peak Learning Rate | 0.1 | 0.1 | 0.1 | 0.1 |
| Weight Decay | $1e-5$ | $1e-5$ | $1e-5$ | $1e-5$ |

Table 8: Hyperparameters for SPARC-GCN

| Parameter | Cora | Citeseer | Pubmed | Reddit |
|---|---|---|---|---|
| Dropout | 0.1 | 0.1 | 0.1 | 0.1 |
| Hidden Dimension | 64, 256, 7 | 64, 256, 6 | 64,256, 3 | 64, 256, 41 |
| Peak Learning Rate | 0.1 | 0.1 | 0.1 | 0.1 |
| Weight Decay | $1e-5$ | $1e-5$ | $1e-5$ | $1e-5$ |

Table 9: Hyperparameters for SPARCphormer

| Parameter | Cora | Citeseer | Pubmed | Reddit |
|---|---|---|---|---|
| Dropout | 0.1 | 0.1 | 0.1 | 0.1 |
| Hidden Dimension | 512 | 512 | 512 | 512 |
| Token List Size | 5 | 7 | 10 | 13 |
| Number of Heads | 8 | 8 | 8 | 8 |
| Peak Learning Rate | 0.001 | 0.001 | 0.001 | 0.001 |
| Weight Decay | $1e-5$ | $1e-5$ | $1e-5$ | $1e-5$ |

## G  OS AND HARDWARE

The training procedures were executed on Rocky Linux 9.3, utilizing Nvidia 578 GPUs including GeForce GTX 1080 Ti and A100 80GB PCIe.

