# OpenReview forum: "G-SPARC: SPectral ARchitectures tackling the Cold-start problem in Graphs"
_ICLR.cc/2025/Conference — ICLR 2025 Conference Withdrawn Submission_

### Official Review · Reviewer_gwDr · 2024-10-29

**Soundness:** 2
**Presentation:** 3
**Contribution:** 1
**Rating:** 3
**Confidence:** 5

**Summary:**

This work studies on the cold-start problem in graph learning and proposes a new method with utilizing spectral embeddings. However, this work also has some significant limitations on motivations and experiments. As such, I give a negative point.

**Strengths:**

1. This work studies on an interesting problem.

2. The proposed method can be integrated into multiple existing methods.

3. Extensive experiments on real-world datasets are conducted to validate the effectiveness of the proposed method.

4. The paper is well-writing.

**Weaknesses:**

1. The motivation of the proposed method is unclear.

a) While this work addresses an interesting problem, there appear to be established techniques that can tackle it. For example, graph structure learning [a1], which refines node relations based on feature similarity and original graph topologies, can significantly enhance the performance of nodes with limited connections. Moreover, graph transformers can effectively learn cold-start node representation based on their feature similarity with other nodes. It remains unclear why these methods cannot adequately address this problem and how the proposed method is superior to them.

b) The rationale for leveraging spectral embeddings is unclear. Why map node features into spectral space rather than the conventional node representation space?

2.  While the authors claim that their method can be integrated into state-of-the-art methods, they only provide examples of Spectral-GCN and NAGphormer, which are relatively outdated. It would be more compelling to provide examples with more recent state-of-the-art methods, such as Difformer [a2].




3. There are also concerns regarding the experiments:

a) The key comparison methods are G-SAGE and C-BREW, but the authors do not provide introductions to these methods. Moreover, more state-of-the-art methods should be included.

b) From Table 6, it is evident that while the integration of SPARC improves the cold-start performance, it significantly degrades the overall performance of the model (sometimes by over 10%). This substantial performance degradation makes the proposed method less applicable in practice.




[a1] WWW24: Self-Guided Robust Graph Structure Refinement

[a2] ICLR’23: DIFFormer: Scalable (Graph) Transformers Induced by Energy Constrained Diffusion

**Questions:**

Please refer to weaknesses.

---

> ### Author Response · Authors · 2024-11-25
> **Authors' response**
>
> Thank you for recognizing the significance of our work, the versatility of our proposed method, the rigor of our experiments, and the clarity of our writing. We also sincerely appreciate the insightful questions you raised.
> Below are our detailed responses to your questions:
>
> Q1a) We appreciate the suggestion to compare with graph structure learning methods like [a1] and have reviewed the work. While graph structure learning can enhance nodes with limited connections, it assumes some connectivity information in addition to feature similarity, which does not address the cold-start problem we focus on—nodes with no connectivity information. Our work (Line 495) shows the limitations of feature-based methods that ignore adjacency, as seen in Figure 6. The results indicate that relying solely on features for predictions leads to performance variability across datasets, making this approach unsuitable for addressing the cold-start node problem in datasets where graph structure plays a larger role.
>
> Q1b)  The use of spectral embeddings is rooted in well-established spectral theory, which is mathematically designed to capture the underlying graph structure. Rather than learning arbitrary node representations, spectral embedding enforces a direct connection to the graph's intrinsic structure. Our framework ensures that the embeddings are meaningful to capture the graph structure and also suitable for various tasks such as node classification, node clustering, and link prediction when integrated into a graph learning method.
>
> Q2) Our current focus is on showcasing how the SPARC framework integrates with two graph learning models—Spectral-GCN and NAGphormer. These examples highlight the flexibility of SPARC, which integrates with existing architectures.
> SPARC is designed to be model-agnostic, meaning its spectral embedding can be easily incorporated into any graph learning method. The core innovation of our framework is its ability to transition nodes into a spectral embedding that captures the graph structure, allowing generalization to cold-start nodes.  This embedding is directly applicable for predicting the neighborhoods of cold-start nodes, which can then be used for all graph learning methods that use the node neighbors.
> While our current experiments focus on Spectral-GCN and NAGphormer, the SPARC embedding can be effortlessly integrated into other methods like Difformer, as well as future advances in graph learning.
>
> Q3a) We acknowledge this gap and agree that including more baselines specifically tailored to cold-start scenarios would strengthen our comparisons. However, we found that such baselines are surprisingly scarce in the literature. This observation further underscores the novelty and importance of our contribution to this underexplored area of graph learning.
> In addition, G-SAGE and C-BREW are introduced in the related work section.
>
> Q3b) Thank you for pointing out the results in Table 6. The observed degradation in performance for non-cold-start nodes is a consequence of the experimental setup. Specifically, the results reflect the accuracy of node classification on non-cold-start nodes when the model is trained on an incomplete graph (i.e., excluding cold-start node links during training).

---

### Official Review · Reviewer_rqyN · 2024-11-02

**Soundness:** 3
**Presentation:** 2
**Contribution:** 2
**Rating:** 3
**Confidence:** 4

**Summary:**

This paper proposes a spectral-based framework that addresses the cold-start problem in graph learning. The key is to leverage spectral embeddings to incorporate cold-start nodes (nodes without any connections to existing nodes). The authors introduce two extensions for GCN and NAGphormer: SPARC-GCN and SPARCphormer to handle cold-start nodes. The approach maps node features to spectral embeddings, enabling the identification of relevant neighbors for cold-start nodes based on the manifold structure of the graph. The authors also demonstrate the applicability of their method to other tasks like node clustering, link prediction, and mini-batching.

**Strengths:**

S1: good idea with spectral embedding for cold-start node in graph learning
S2: Extend Spectral-GCNs and Graph Transformers to support cold-start nodes while retaining their performance on connected nodes.
S3: multiple tasks: node classification, node clustering, link prediction, and mini-batching.

**Weaknesses:**

C1: How does the proposed method handle scenarios where node features are not informative, e.g., in recommendation, cold-start users are initialized with random feature or mean values of each feature?

C2: Can the spectral embedding approach be extended to handle dynamic graphs, where new edges are constantly added or removed?

C3: Computational complexity: the proposed vs existing graph learning techniques, especially for large-scale graphs. For example, in Figure 4, Metis partitioning is a classic fast method. Simply comparing convergence iteration is not convincing enough. What about running time? How about the complexity of this special-space related extension vs original algorithm GCN and NAGphomer?

C4: How can the authors cite an under-review paper in the same conference ICLR25?  "Anonymous. Grease: Generalizable spectral embedding with an application to umap. Under review ICLR 2024, attached to supplementary, 2024" In fact, by checking the sup doc, this draft is under review for ICLR25 instead of ICLR24.

C5: Lack of baselines specifically designed for cold-start nodes or dynamic graph scenarios?

**Questions:**

Please see weakness.

---

> ### Author Response · Authors · 2024-11-25
> **Authors' response**
>
> We thank the reviewer for their thoughtful and detailed feedback. We appreciate their recognition of the strengths of our method and their valuable suggestions, which have helped us identify areas for further clarification and improvement.
>
> C1.The scenario described, where node features are random or based on mean values, is different from the type of cold-start problem our method addresses. Our focus is on nodes with meaningful features but no connections, which is a common setting in graph learning.
> Also in recommendation systems, the cold-start problem often involves new items or users that come with associated information but lack connections, rather than nodes initialized with entirely uninformative features.
>
> C2. This is an excellent and highly relevant point, particularly in the context of the cold-start scenario, where graph structures are often subject to dynamic changes. While our proposed method is not explicitly designed to handle dynamic graphs in its current form, we recognize the importance of this challenge and its relevance to real-world applications.
> One potential approach to extending our method for dynamic graphs is to leverage the evolving graph structure—specifically, newly added or removed edges—as an additional signal for learning. This can be achieved by incorporating a dynamic link prediction loss into the finding of /mathcal{F}. The idea would be to use the spectral embeddings produced by our method to predict the presence of edges and compute a loss by comparing these predictions with the actual changes in the graph structure. This additional loss would allow the model to adapt its spectral embeddings in response to dynamic changes, potentially enhancing its robustness to evolving graph properties.
> Although this direction is beyond the scope of our current work, it represents an exciting avenue for future research, with the potential to address dynamic graph learning and strengthen the applicability of spectral methods in such scenarios. We appreciate this insightful observation and believe it highlights an important opportunity for further development.
>
> C3. In the current version, we primarily focus on convergence behavior as a measure of computational efficiency.  While the spectral embedding computation involves an eigen-decomposition step, which can be computationally intensive, we mitigate this through efficient approximations, particularly for large graphs. We will ensure that these trade-offs are explicitly detailed in the final version of the paper.
> In Figure 4, we isolated the graph to evaluate the "quality" of the mini-batches, which aligns with our intention to continue exploring this approach for parallel training.
> Furthermore, our parametric map that learns the spectral embedding of nodes is not only generalizable but also faster than other eigendecomposition techniques. Both models, Spectral-GCN and NAGphormer, use eigendecomposition in their architectures: Spectral GCN employs the Laplace operator, while NAGphormer utilizes positional encoding, making the preprocesses comparable across the datasets used.
>
> C4. Thank you for pointing this out. We appreciate your careful attention to detail and for bringing this to our attention. This paper will be publicly published soon, and we’ll change the citation accordingly.
>
> C5. We acknowledge this gap and agree that including more baselines specifically tailored to cold-start scenarios would strengthen our comparisons. However, we found that such baselines are surprisingly scarce in the literature. This observation further underscores the novelty and importance of our contribution to this underexplored area of graph learning.

---

### Official Review · Reviewer_LDCT · 2024-11-04

**Soundness:** 3
**Presentation:** 3
**Contribution:** 2
**Rating:** 5
**Confidence:** 4

**Summary:**

This paper proposes a spectral-based method to generalize the graph learning to handle cold start nodes. In particular, the authors propose to introduce node features into SOTA graph learning models and combine it with the abstracted graph structure, e.g., eigenvectors of the Laplacian matrix, to handle cold-start nodes. Through experiments, the authors demonstrate the effectiveness of the proposed method.

**Strengths:**

1. Overall, the paper is well written and easy to follow.
2. The motivation of this work is well presented. The literature is well surveyed and discussed.
3. The proposed method is well elaborated, and the examples of integrating this method into SOTA graph learning methods are described clearly.
4. Experiments are conducted on several datasets to showcase the effectiveness of the proposed method.

**Weaknesses:**

1. It's unclear how this proposed method can be used to handle directed graph, e.g. follow-relationship in social networks. Eigenvectors of the Laplacian matrix is suitable for undirected graph, how can we use it to capture the asymmetric relationship between nodes is not discussed.
2. There is no discussion or experiment to discuss how robust the proposed method is towards the sparsity of the graph. As pointed out by the authors, graphs are constantly evolving, it's unclear how the existing graph structure can better capture the underlying overall graph property and how it can capture the change dynamically.
3. There is no feature-based baseline comparison to validate the improvement of using graph structure.

**Questions:**

1. How will the proposed method capture the directed and asymmetric relationship between nodes, which is quite common in real world applications?
2. How is the model performance when we have different percentage of cold-start nodes? Taking Reddit dataset as an example, how does the proposed method perform when we have 10%, 20%, 30%, 50% nodes as cold-start nodes?
3. How does this proposed method compare with the pure feature-based methods for node classification and clustering? E.g. using transformer to encode the node features.

---

> ### Author Response · Authors · 2024-11-25
> **Authors' response**
>
> Thank you for your thoughtful review, we appreciate you found our paper clear, with strong motivation, and comprehensive experiments. We also appreciate your constructive feedback these insights are invaluable for improving our approach.
> Below are our detailed responses to your questions:
>
> Q1. Handling directed graphs and asymmetric relationships is undoubtedly important, but it is not a common requirement in graph learning. A significant portion of research in this field focuses on undirected graphs, which are both highly relevant and widely studied across various domains. Our proposed method aligns with the majority of the papers in graph learning.
>
> Q2. As the percentage of cold-start nodes is very high, such as 50%, the graph structure becomes significantly sparse. In such scenarios, the connectivity within the graph diminishes, leading to a substantial loss of structural information. Consequently, the graph effectively transitions from a structure-based representation to a feature-based one, with limited connections between nodes to propagate meaningful information.
> It is important to note that in real-world cold-start scenarios, the percentage of cold-start nodes is typically not this large. For instance, in social media platforms, where cold-start problems are prominent, the fraction of new users introduced at any time is often much smaller.
> Our experiments and analysis in the paper primarily focus on cold-start percentages that align with these realistic scenarios, demonstrating the effectiveness of the proposed method under practical conditions.
>
> Q3. In the paper, we present experiments evaluating feature-based methods that completely ignore graph structure (see Figure 6). The results indicate that relying solely on features for predictions leads to performance variability across datasets, making this approach unsuitable for addressing the cold-start node problem.

---

### Author Response · Authors · 2024-11-25
**Official Comment by Authors**

We sincerely thank you for the time and effort you have dedicated to reviewing our paper. Your feedback is invaluable and has greatly helped us to better understand the strengths and weaknesses of our work. We are grateful for your thoughtful comments and suggestions, which have already contributed to improving the quality of our paper.

---

### Note · Authors · 2024-12-02

**Comment:**

We thank the reviewers for their comments and feedback. Based on the current comments, we have decided to withdraw this version.

**Withdrawal Confirmation:**

I have read and agree with the venue's withdrawal policy on behalf of myself and my co-authors.